# Evaluation of the Effect of the *Mycoplasma hyopneumoniae* Live Vaccine (Strain 168) in Ningxiang Pigs

**DOI:** 10.3390/vaccines12121332

**Published:** 2024-11-27

**Authors:** Zhanguo Qin, Pengfei Zhao, Lunyong Chen, Zhen Han, Yuankui Zhang, Junlong Zhao

**Affiliations:** 1National Key Laboratory of Agricultural Microbiology, College of Veterinary Medicine, Huazhong Agricultural University, Wuhan 430070, China; zhanguoqin@webmail.hzau.edu.cn (Z.Q.); pengfeizhao@webmail.hzau.edu.cn (P.Z.); chenlunyong@webmail.hzau.edu.cn (L.C.); hanzhen@webmail.hzau.edu.cn (Z.H.); 2Zhaofenghua Group Beijing Research Institute, Beijing 102600, China; zhang20130123@outlook.com

**Keywords:** *Mycoplasma hyopneumoniae* live vaccine (strain 168), lung lesion score at slaughter, *pneumonia* reduction rate, SIgA and IgG antibodies

## Abstract

[Background/Objectives] *Mycoplasma hyopneumoniae* (*M. hyopneumoniae*) is widespread in the global swine industry, leading to significant economic losses, and is particularly severe in native Chinese pig breeds. The Ningxiang pig, a well-known native breed in China, is susceptible to *M. hyopneumoniae*, exhibiting high morbidity and mortality rates. This study was designed to evaluate the clinical effectiveness of the *M. hyopneumoniae* live vaccine (strain 168). [Methods] The vaccine was delivered to 7-day-old piglets in the farrowing room through intrapulmonary administration, and its efficacy was compared with that of the *M. hyopneumoniae* inactivated vaccine (strain J). Four experimental groups were designed: Group 1 (inactivated vaccine + inactivated vaccine), Group 2 (live vaccine + inactivated vaccine), Group 3 (live vaccine), and Group 4 (control), which was not vaccinated. The production performance of each group was measured, and the lung lesion scores and pneumonia lesion reduction rates were evaluated at slaughter. Nasal swabs and serum samples were collected on days 0, 14, 28, 56, 84, 112, and 140 to assess SIgA, IgG antibody levels, and the *M. hyopneumoniae* pathogen. [Results] The results showed that Group 3 had the best production performance and clinical outcomes, with the lowest average lung lesion score, of 4.43 ± 2.44, which was significantly different from the other groups (*** *p* < 0.001). [Conclusions] This study provided scientific evidence to support vaccination strategies for preventing and controlling the *M. hyopneumoniae* in native pig populations.

## 1. Introduction

Respiratory diseases of pigs have significantly impacted swine populations worldwide; *M. hyopneumoniae* is recognized as one of the most critical primary bacterial respiratory pathogens [1]. This disease is widespread in the global swine industry, leading to significant economic losses. The primary clinical symptoms include chronic coughing, reduced average daily weight gain (ADG), and decreased feed conversion efficiency [2]. This pathogen often co-infects with other respiratory diseases in pigs, leading to the development of Porcine Respiratory Disease Complex (PRDC) [3]. Common bacterial co-infections include *Actinobacillus pleuropneumonia* (APP) and swine pasteurellosis. Viral co-infections often involve Porcine Circovirus (PCV2 and PCV3) [4], Porcine Reproductive and Respiratory Syndrome (PRRS), and Swine Influenza Virus (SIV) [2].

There has been growing international attention on controlling *M. hyopneumoniae* infections in pigs in recent years. Vaccination is widely used globally to control *M. hyopneumoniae* infections in pigs. The commercial vaccines available primarily include inactivated and attenuated live vaccines, formulated using adjuvants combined with either inactivated whole bacteria or attenuated live bacteria (such as strain 168) [5,6]. Inactivated vaccines primarily induce humoral immunity, while attenuated live vaccines generate humoral immunity, and the host stimulates mucosal and cellular immunity. When live attenuated *M. hyopneumoniae* vaccines enter the host, they create an early occupancy effect in the lungs, preventing the proliferation and infection of wild-type strains, thereby providing adequate piglet protection [7].

The Ningxiang pig, a native breed from the Hunan province of China, is particularly susceptible to *M. hyopneumoniae*, with relatively high morbidity and mortality rates. To address the challenges posed by outbreaks of enzootic pneumonia and persistent mortality due to *Mycoplasma* infections in Ningxiang pigs, the *M. hyopneumoniae* live vaccine (strain 168) was administered via intrapulmonary-targeted immunization to 7-day-old piglets in the farrowing room for early immunization in this study. The inactivated vaccine primarily induces humoral immunity, while the attenuated live vaccine mainly induces cellular and mucosal immunity [8]. Analysis suggests that the “occupancy effect” could be one of the unique immunological mechanisms of the attenuated *M. hyopneumoniae* live vaccine. The occupancy effect is a distinct immune mechanism of attenuated live vaccines, and the *M. hyopneumoniae* live vaccine might prevent infection by field strains through this mechanism. It is hypothesized that the attenuated live *M. hyopneumoniae* vaccine may take advantage of the window in which piglets do not naturally become infected with *M. hyopneumoniae*, 5–7 days after birth. The intrapulmonary vaccine directly reaches the alveolar mucosal surface, forming a dominant occupancy on the alveolar cells. The vaccine strain proliferates rapidly using nutrients such as mucosal proteins, eventually occupying all the available space on the alveolar surface (which may take about 1–2 weeks). At this time, if the piglets inhale and are exposed to the wild-type *M. hyopneumoniae*, the alveolar surface will lack space for the wild strain to adhere, thereby reducing the occurrence of *M. hyopneumoniae* infection [9,10].

The effectiveness of this approach was compared with that of an imported commercial inactivated *M. hyopneumoniae* vaccine (strainJ) administered via intramuscular injection. Specifically, it includes a comprehensive analysis of the pigs’ SIgA and IgG antibody levels, antigens, production performance indicators (such as average daily gain and mortality rate), and lung lesion scores. This comparison aimed to comprehensively evaluate the protective efficacy and immunization schedules of different *M. hyopneumoniae* vaccines in Ningxiang pigs, providing new strategies and data references for the clinical prevention and control of *M. hyopneumoniae* in native pig breeds. Ningxiang pigs are one of the four most renowned pig breeds in China and among the first local pig breeds in the country to have a national standard. The breed has also been designated by the Ministry of Agriculture as one of the first national-level genetic resource conservation breeds in poultry and livestock. *Mycoplasma pneumonia* in swine severely harms native pig breeds and causes significant economic losses. For the first time, this study systematically explores the immunological prevention and control of *M. hyopneumoniae* in native pig breeds, providing new approaches for the prevention and control of *Mycoplasma pneumonia* in swine in these breeds.

## 2. Materials and Methods

### 2.1. M. hyopneumoniae Vaccine and Experimental Animals

The commercial *M. hyopneumoniae* live vaccine (strain 168), with an antigen content of ≥10^6^ CCU per dose, was obtained from Zhaofenghua Biological (Nanjing) Technology Co., Ltd., Nanjing, China. The *M. hyopneumoniae* inactivated vaccine (strain J), with an antigen content of ≥10^5.5^ TCID_50_ per dose, was sourced from HIPRA, Spain. A total of 313 male and female Ningxiang piglets, aged 7 days, were purchased from a breeding farm in Hunan; all tested negative for antigens (PCR amplification) and antibodies (ELISA) against African swine fever, porcine reproductive and respiratory syndrome virus, classical swine fever virus, and *pseudorabies* virus, and were *M. hyopneumoniae* antibody-positive and pathogen-negative.

### 2.2. Animal Experiment

The experimental piglets were randomly divided into four groups: Group 1 (two immunizations, inactivated vaccine + inactivated vaccine) with 98 piglets; Group 2 (two immunizations, live vaccine + inactivated vaccine) with 98 piglets; Group 3 (one immunization, live vaccine) with 96 piglets; and a blank control group, Group 4, with 21 piglets. The *M. hyopneumoniae* inactivated vaccine was administered intramuscularly, while the attenuated live vaccine was administered via intrapulmonary injection. The pigs in Groups 1 and 2 received their first immunization at seven days old and a second immunization at twenty-one days old, whereas those in Group 3 received a single immunization at seven days old (Table 1). The experiment started at 7 days of age and ended at 230 days of age. The trial was conducted at the Hua Pig Farm in Ningxiang, Hunan Province. All animal procedures adhered to the relevant animal welfare regulations and usage guidelines of the China Experimental Animal Center and met the standards set by the Ethics Committee of Huazhong Agricultural University (HZAUSW-2022-0007).

### 2.3. Sample Collection and Testing

Nasal swabs and blood samples were collected from the experimental pigs at 0, 14, 28, 56, 84, 112, and 140 days after the start of the experiment. These samples were then tested for the presence of the *M. hyopneumoniae* pathogen, SIgA through a nasal swab, and serum IgG antibodies. All samples were properly labeled, stored and transported at 4 °C, and delivered to the laboratory within 24–36 h. The SIgA in the nasal swabs was measured using a commercial SIgA antibody detection kit (SIgA-ELISA, Veterinary Research Institute, Jiangsu Academy of Agricultural Sciences, Nanjing, China) [11]. The specific procedure is as follows: All reagents were brought to room temperature (15–25 °C) before use. A total of 100 µL each of the negative control, the positive control, and the test samples were added to the antigen-coated plates and incubated at 37 °C for 120 min. After incubation, the liquid was discarded, and each well was washed five times with 250 µL of wash buffer. Next, 100 µL of HRP-conjugated anti-pig IgA was added to each well and incubated at 37 °C for 60 min. The washing steps were then repeated, and 100 µL of the substrate solution was added to each well, followed by a 7 min incubation at 37 °C in the dark. Finally, 50 µL of stop solution was added to each well, and the absorbance was read at 450 nm using a microplate reader. The S/P value was calculated as follows: S/P = (OD450 nm of the sample − OD450 nm average of the negative control)/(OD450 nm average of the positive control − OD450 nm average of the negative control). An S/P value < 0.15 was considered negative, >0.20 was considered positive, and values between 0.15 and 0.20 were considered doubtful.

The IgG antibodies in pig serum were detected using an *M. hyopneumoniae* Antibody Test Kit (AsurDxTM, Dallas, TX, USA), as follows: First, the test samples were diluted at a 1:5 ratio using the sample diluent. Then, 90 µL of the MHP detection solution was added to each well of the antigen-coated plate, followed by the addition of 10 µL each of the positive control, the negative control, and the diluted samples. The plate was incubated at 20–25 °C for 30 min. After incubation, the liquid was discarded, and each well was washed five times with 250 µL of wash buffer. Next, 100 µL of the enzyme-labeled antibody was added to each well, and the plate was incubated in the dark at 20–25 °C for 30 min. The washing process was repeated, and 100 µL of TMB was added to each well, followed by incubation in the dark at room temperature for 15 min. Finally, 100 µL of stop solution was added to each well, and the absorbance was read at 450 nm using a microplate reader. The PP value for each sample was calculated as follows: PP value = (OD value of the sample/OD value of the positive control) × 100%. A sample PP value of less than 30% indicated a negative antibody result, while a PP value greater than 30% indicated a positive antibody result. For detecting Mhp pathogens, real-time quantitative PCR (qPCR) was used, following the protocol provided with the Mhp qPCR detection kit. Amplification efficiency should be determined from the slope of the log-linear portion of the calibration curve. Specifically, PCR efficiency^−1/slope^ − 1, where the logarithm of the initial template concentration (the independent variable) is plotted on the x axis and Cq (the dependent variable) is plotted on the y axis. The theoretical maximum of 1.00 (or 100%) indicates that the amount of product doubles with each cycle [12]. DNA was extracted from the samples using a magnetic bead extraction method. The PCR reaction mixture was prepared with a total volume of 20 μL, including 15 μL of the PCR reaction solution, 2 μL of the P solution, and 3 μL of the DNA sample. The PCR cycling program was as follows: 95 °C for 3 min (initial denaturation), 45 cycles at 95 °C for 10 s, and 60 °C for 30 s. Interpretation: if the sample Ct value was ≤38, the result was considered positive, otherwise, it was considered negative. Based on the *M. hyopneumoniae*-183 gene sequence of Mhp recorded in GenBank, specific primers and probes were designed for the conserved region using Primer-BLAST (Table 2). The primers and probes were synthesized by Sangon Biotech (Shanghai) Co., Ltd. (Shanghai, China).

### 2.4. Production Performance and Clinical Observations

The birth weight, weaning weight, transfer weight (28 days), and market age weight were recorded for each experimental pig. The ADG for the nursery phase in the experimental groups was calculated, excluding the data from deceased pigs in the performance evaluation. Mortality rate data for piglets in the farrowing and nursery phases, weaning weight in the farrowing room, ADG during the nursery phase, and treatment plans. ADG was calculated as (final weight − initial weight)/number of days measured. The mortality rate was calculated as (number of deaths/number of pigs raised) × 100%.

### 2.5. Lung Lobe Lesion Index Statistics and Lung Scoring at Slaughter

Forty pigs (10 pigs from each group), aged 230 days, were randomly selected and sent to a designated slaughterhouse for observation of lung lesions. Lung lesions were evaluated and recorded using the “28-point scoring system for *M. hyopneumoniae* lung lesions” [13]. A lesion score of more than 10 points was classified as diseased, while a score of less than 4 points indicated health. All scoring was performed by the same individual, who was a specialized scientist blinded to the experimental arrangements. Additionally, the total number of affected lungs, the average lesion score of the affected lungs, the overall average lesion score, and the pneumonia reduction rate were recorded. The calculations were as follows: percentage of affected lungs = (number of affected lungs/total observed) × 100%; average lesion score of affected lungs = total lesion scores of affected lungs/number of affected lungs; overall average lesion score = total lesion scores of affected lungs/total number of lungs examined; pneumonia lesion reduction rate = 100% × (average lung lesion score of control group − average lung lesion score of vaccinated group)/average lung lesion score of control group [14,15].

### 2.6. Statistical Analysis

The experiment involved recording and carrying out statistical analysis of the weaning weight in the farrowing room, ADG during the nursery phase, the antibody levels of SIgA and IgG at various time points, and lung score data using Excel software (version 2010). The Shapiro–Wilk normality test revealed that the collected data in this study were consistent with a normal distribution. One-way analysis of variance (ANOVA) was used to assess the variance among the experimental groups, followed by Tukey’s test for multiple comparisons between groups with significant differences [16], where *p* < 0.05 was considered statistically significant (*). The analysis and statistical graph plotting were performed using GraphPad Prism 10.1.2 software.

## 3. Results

### 3.1. Medication Usage Results During the Weaning Stage

Before the weaned pigs were transferred to the fattening stage, the medication regimen during the experimental period was recorded and summarized, as shown in Table 3. It can be observed that Group 1 had the highest number and variety of medications, followed by Group 2, with Group 4 having a lower number and variety of medications, and Group 3 having the least number and variety of medications.

### 3.2. Weaning Weight and Mortality Rate of Suckling Piglets in Each Group

At 28 days of age, the piglets in each group were weighed to calculate the average weaning weight, and the mortality rate of the suckling piglets (weaned at 28 days of age) was recorded. The results are detailed in Table 4. The highest average weaning weight was observed in Group 3, at 6.63 ± 0.48 kg, while the lowest was in Group 2, at 5.61 ± 0.48 kg. The mortality rate was 3.1% in Group 1, 1% in Group 2, and 0% in both Group 3 and Group 4.

### 3.3. Average Daily Weight Gain and Mortality Rate During the Nursery Phase for Each Group

As shown in Table 5, the ADG was 0.33 ± 0.05 kg in Group 1, 0.30 ± 0.05 kg in Group 2, 0.35 ± 0.05 kg in Group 3, and 0.34 ± 0.06 kg in the control, Group 4. The highest mortality rate was observed in Group 1 at 4.2%, followed by Group 2 at 2.1%, while the mortality rates in Group 3 and Group 4 were both 0%.

### 3.4. Lung Lesion Scores and Average Pneumonia Reduction Rate at Slaughter

Lung lesion scores at slaughter were used to calculate the *pneumonia* reduction rate for each immunization group and the control group. The difference between Group 3 and Group 2 was not statistically significant (*p* > 0.05), and the difference between Group 1 and Group 4 was also not statistically significant (*p* > 0.05). However, the differences between Group 3 and Group 4, as well as between Group 2 and Group 4, were highly significant (*** *p* < 0.001) (Table 6).

The *pneumonia* lesion reduction rates for Group 1, Group 2, and Group 3 were 10.42%, 57.64%, and 69.44%, respectively (Table 6). The reduction rates were 10.41% for Group 1, 57.64% for Group 2, and 69.44% for Group 3, with Group 3 > Group 2 > Group 1. Considering the effectiveness, convenience, and cost-effectiveness for clinical use, Group 3, administered via intrapulmonary injection, was determined to be the best choice.

### 3.5. Results of the M. hyopneumoniae-Specific SIgA Antibody Detection in Nasal Swabs

Significant differences in the SIgA antibody S/P values were observed only at day 14 post-immunization, with all vaccinated groups showing significantly higher values than the control group. No significant differences were observed at any other time point (Figure 1). Additionally, there was no significant correlation between the SIgA antibody positivity rate and the time post-immunization across the groups.

In Group 1, the average SIgA antibody S/P value peaked at 0.471 at day 14 post-immunization, increased again to 0.396 at day 56, and then gradually declined. In Groups 2 and 3, the average SIgA antibody S/P values gradually increased after immunization. In Group 4, the control group, the highest levels were reached at day 56, followed by a gradual decline (Figure 2, Table 7).

### 3.6. Results of the M. hyopneumoniae-Specific IgG Antibody Detection in Serum

Significant differences in the IgG antibody PP values were observed between the groups only at days 28 and 84 post-immunization, with Group 3 consistently showing significantly lower IgG antibody PP values compared to the other groups (Figure 3). The IgG antibody positivity rate in all groups increased significantly post-immunization, showing a clear positive correlation. Similarly, the average IgG antibody PP values in each group also showed an increasing trend over time post immunization (Figure 4, Table 8).

### 3.7. Results of the M. hyopneumoniae DNA Detection in Nasal Swabs

In Groups 1 and 2, *M. hyopneumoniae* pathogens were detected at days 28 and 140 post immunization, with a positivity rate of 8.33% in both groups. In Groups 3 and 4, the positivity rates for the *M. hyopneumoniae* pathogens were higher. Specifically, in Group 3, the pathogen was detected at days 14 (25%), 112 (66.7%), and 140 (25%) post immunization. In Group 4, *M. hyopneumoniae* was detected at all time points except on day 0, day 84, and day 140 post immunization, when the swabs were negative (Figure 5, Table 9).

## 4. Discussion

Infection with *M. hyopneumoniae* in pigs leads to reduced ADG, increased mortality, and higher medication costs [17,18,19]. However, there are still quite a few studies on whether the inactivated vaccine immunization can reduce infection and then affect the average weight gain of pigs [20]. However, there are relatively few studies on live vaccine immunization for this purpose [9]. Attenuated vaccines against *M. hyopneumoniae* have been licensed in Mexico and in China [21]. The vaccine in Mexico is a thermosensitive mutant of *M. hyopneumoniae* (strain LKR) (VaxSafe^®^ MHYO, AviMex) that should be applied intranasally once from 3 days of age onwards. The attenuated Chinese vaccine strain is derived from a virulent parent strain 168, isolated in 1974 from an Er-Hua-nian pig (Chinese native breed, very sensitive to *M. hyopneumoniae*) with typical signs and lesions of enzootic pneumonia [22,23]. At 7 days of age, piglets were injected perpendicularly between the ribs, 2 cm behind the right scapula, with one dose per piglet.

Maes et al. conducted a study on the effect of Mhp vaccination on ADG under different commercial pig production conditions. They found that from day 8 after birth until slaughter, vaccinated pigs gained 14 g more per day compared to control pigs, and during the growing/fattening period, they gained 25 g more per day [24,25]. Villarreal et al. (2011) conducted an experiment on a pig farm with porcine mycoplasma infection. They immunized weaned piglets with a commercial vaccine at 3 weeks of age and then measured the ADG at different growth stages. The ADG of the immunized group at 3–9 weeks of age was 70.7 g/head/day (24%) less than that of the control group, and the difference was extremely significant (** *p* < 0.01) [26]. In this study, during the nursery phase, the ADG was 0.33 ± 0.05 kg in Group 1, 0.30 ± 0.05 kg in Group 2, 0.35 ± 0.05 kg in Group 3, and 0.34 ± 0.06 kg in the control, Group 4. There was no significant difference among Groups 2, 3, and 4; however, these three groups differed significantly from Group 1 (* *p* < 0.05). The mortality rate of Group 1 was the highest, at 4.2%. The results of this study are not consistent with the trend of the above-mentioned research findings. The possible reason is that this may be related not only to the good protective effect of early-age live vaccine immunization but also to the stress response caused by vaccination. The stress response from the second immunization is greater than that from the first, thereby affecting ADG.

At the 28-day weaning stage in the farrowing room, the weaning weights of the piglets in Groups 1, 2, 3, and 4 were 5.76 ± 0.51, 5.61 ± 0.48, 6.63 ± 0.48, and 6.50 ± 0.65 kg, respectively. Group 3 showed a significant difference compared to the other three groups (* *p* < 0.05). The weaning mortality rates for Groups 1, 2, 3, and 4 were 3.1%, 1.1%, 0%, and 0%, respectively. Possible reasons include the susceptibility of local Chinese pig breeds to *Mycoplasma* infection, which may lead to early infection in the farrowing room, resulting in a reduced weaning weight or even mortality. Another factor could be the health status of the newborn piglets, as piglets with lower health tend to have slower growth and a lower weaning weight. Vaccine-related immune stress may also have contributed to these results.

The lung lesion scores of fattening pigs can be used as an evaluation method based on the visual examination of organ tissues for lesions in market-ready pigs [27,28]. This method is crucial for monitoring the health status of the herd and serves as an important tool for assessing disease resistance and vaccine efficacy in pigs [29]. In recent years, researchers have investigated whether vaccination can reduce the lung damage caused by *Mycoplasma* infection [30]. Sibila et al. (2007) demonstrated that administering a double-dose vaccine at 1 and 3 weeks of age effectively reduced the prevalence of *Mycoplasma-like* lung lesions at slaughter compared to the control group [31]. In another investigation by Meyns et al. (2006), 30 piglets received a vaccination at 1 week of age, while an additional 30 unvaccinated piglets, confirmed to be free of *M. hyopneumoniae*, were allocated into six separate pens. Within each pen, three animals were inoculated intratracheally with *M. hyopneumoniae* and housed alongside seven unexposed piglets during the standard 6-week nursery phase. The findings revealed that vaccinated contact piglets showed a markedly lower average lung lesion score of 0.18 compared to the significantly higher score of 1.95 observed in their non-vaccinated counterparts (* *p* < 0.05) [32]. In this study, the average lung lesion scores in Groups 1, 2, 3, and 4 were 12.90 ± 4.28, 6.78 ± 4.32, 4.43 ± 2.44, and 14.40 ± 3.53, respectively. The difference between Group 3, Group 1, and Group 4 is extremely significant (*** *p* < 0.001), and the difference between Group 3 and Group 2 is not significant (*p* > 0.05). Compared with the control group, the immune groups are different. Vaccination can reduce post-slaughter lung lesions to a certain extent. The results of this study are consistent with the results of the above studies.

The respiratory tissues in the lungs are the most vulnerable areas to pathogen attacks in animals. Numerous studies have shown that SIgA is widely distributed on the outer surfaces of the bronchial tree at all levels and in the lung parenchyma. The SIgA immune response is a crucial defense mechanism of the bronchial tree and the lung *parenchyma* against infections [33]. Therefore, when evaluating a vaccine’s protective efficacy on mucosal tissues, the secretion level of SIgA is a better indicator of the overall mucosal immune response. On the other hand, serum antibody IgG is considered by Renegar as the backup force for the SIgA mucosal defense system [34]. This approach is commonly used as a standard method for diagnosing *M. hyopneumoniae* infection and evaluating vaccine efficacy. Therefore, to accurately assess the effectiveness of respiratory immunity, the detection of SIgA antibodies in the lungs should be combined with the evaluation of specific serum IgG antibodies. This combined assessment more accurately reflects the actual protective effect of the vaccine against *M. hyopneumoniae* infection [11]. In this study, we conducted a comprehensive evaluation of vaccine efficacy by measuring both SIgA and IgG antibody levels, consistent with that outlined in the literature [35,36]. Feng et al. (2010) examined the immune response triggered by the attenuated *M. hyopneumoniae* 168 strain vaccine. Piglets between 8 and 15 days old were immunized through the intrapulmonary route, and both serum-specific IgG antibodies and SIgA antibodies in bronchoalveolar lavage fluid were analyzed at 30 and 60 days post immunization (DPI), respectively. The findings indicated a rapid elevation of SIgA levels in the respiratory tracts of vaccinated pigs, reaching a maximum at 60 DPI. However, no detectable levels of serum IgG antibodies against *M. hyopneumoniae* were observed during the entire study period [22]. Additionally, ELISA analysis showed that the nasal swab SIgA antibody S/P values exhibited noticeable differences among the pig groups as early as 14 days following vaccination. The vaccine-immunized groups were significantly higher than the blank control group (* *p* < 0.05), and there were no significant differences in the other time periods. There was no significant correlation between the positive rate of SIgA antibodies to *M. hyopneumoniae* in each group and the time after immunization. Group 1 may be infected 14 days after immunization, and Groups 2 and 3 have different degrees of SIgA antibody positivity, which may be related to the immunization of live vaccines. The immune response of live vaccines is mainly cellular immunity and mucosal immunity. After immunization with live vaccines, the SIgA antibodies produced by mucosal immunity in Groups 2 and 3 reached a peak 56 days after immunization, which is consistent with the results of Feng et al. (2010). The results of the ELISA test showed that there were significant differences in the average IgG antibody PP values among the groups 28 and 84 days after serum-specific antibody IgG immunization (* *p* < 0.05), and the average IgG antibody PP value of Group 3 was significantly lower than that of other groups (* *p* < 0.05). This is consistent with the results of Feng et al. (2010).

Like many other pathogens, *M. hyopneumoniae* infection and vaccination can induce specific immune responses in animals, with specific antibody levels serving as indicators for evaluating infection or vaccine efficacy. However, there has been a lack of reliable serological methods to assess the protective efficacy of *M. hyopneumoniae* vaccines, particularly live vaccines. It is generally believed that there is no direct correlation between serum antibodies and immune protection; for *M. hyopneumoniae* infection of pigs, no data on antibody transfer have been published, but data published in 1969 would indicate that the antibodies are not protective. In fact, following vaccination, no correlation was observed between the antibody levels and the reduction in the number of lung lesions and no protection of piglets was shown from the vaccinated sows [37]. This was confirmed later by others [35,38]. In addition, a longitudinal study including 825 pigs in eight chronically infected herds identified no overall relationship between the average daily weight gain and the serological response to *M. hyopneumoniae* [39]. Furthermore, the seroconversion rates in pigs vary widely after vaccination, ranging from 30% to 100%. These variations may be attributed to differences in vaccine strains, adjuvant types, and detection methods [40].

Researchers have conducted extensive studies on whether vaccination can prevent *M. pneumonia* swine infection. Baccaro et al. (2006) found that, before immunization, there was no significant difference in the percentage of *Mycoplasma* pathogen detected in tonsil swabs via PCR between the two vaccine groups and the control group. The detection rates were 51.9% and 29.6% in the two vaccine groups and 50% in the control group. At days 35, 66, and 97 post immunization, there were no significant differences among the groups, and by day 97, the detection rates were high across all groups, being 92.0% and 92.3% in the vaccine groups and 92.6% in the control group. However, at day 125 post immunization, there was a highly significant difference between the vaccine groups and the control group, with detection rates of 52.0% and 65.4% in the vaccine groups compared to 92.6% in the control group [40]. It can be seen that the inactivated vaccine failed to prevent *M. hyopneumoniae* infection. Meyns et al. (2006) compared the effects of vaccination on MPS infection in pigs under experimental conditions and found that vaccination could not prevent the colonization of *M.hyopneumoniae* in the lungs [32].

This study found that the antigen positivity rates in Immunization Group 1, Immunization Group 2, and Immunization Group 3 varied significantly at different time points, with particularly notable fluctuations observed in the control group, Group 4. Our findings are consistent with the results of the studies mentioned above.

The local Ningxiang pig breed generally has a longer feeding period, about 60 days longer than that of imported breeds, with a market-ready age of approximately 250 days. Ningxiang pigs often experience a second infection after reaching 200 days of age. As the age increases, the incidence of disease rises, but the mortality rate remains relatively low. Based on the positive rate of *Mycoplasma* pathogens on this farm, it is proposed that *Mycoplasma* infection may occur at multiple ages, specifically around 20 days and 100 days. From the trend of IgG antibodies, it is inferred that there may be a chance of reinfection after 100 days [32]. Therefore, the control group shows a low mortality rate but a high rate of lung infection.

The main limitation of this study is that the evaluation of the live vaccine (strain 168) was conducted solely on commercial Ningxiang pigs, a Chinese native pig breed, without assessing its efficacy in breeding herds of Ningxiang pigs and evaluating the cellular immune responses in porcine mycoplasmal pneumonia, including monitoring key indicators such as IFN-γ, IL-4, and TNF-α.

## 5. Conclusions

In this study, which focused on the Chinese native Ningxiang pig breed, Group 3 showed the best production performance and clinical outcomes, with the lowest average lung lesion score of 4.43 ± 2.44, which was significantly different from the other groups (*** *p* < 0.001). The control group, Group 4, had the highest average lung lesion score at slaughter, at 14.40 ± 3.53. The *pneumonia* lesion reduction rates were 10.41% for Group 1, 57.64% for Group 2, and 69.44% for Group 3. These results showed that the live vaccine of *M. pneumonia* could be used to prevent porcine enzootic *pneumonia* in Ningxiang native pigs.

## Figures and Tables

**Figure 1 vaccines-12-01332-f001:**
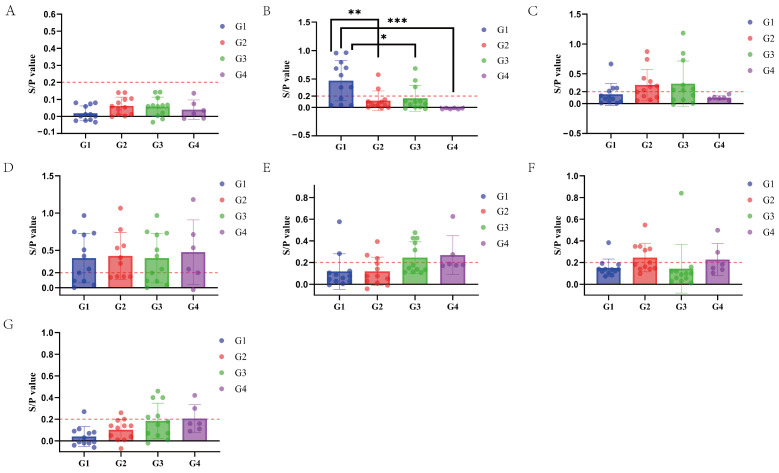
Results of SIgA antibody levels at different ages after immunization. (**A**): SIgA antibody level at day 0 after immunization. (**B**): SIgA antibody level at day 14 after immunization. (**C**): SIgA antibody level at day 28 after immunization. (**D**): SIgA antibody level at day 56 after immunization. (**E**): SIgA antibody level at day 84 after immunization. (**F**): SIgA antibody level at day 112 after immunization. (**G**): SIgA antibody level at day 140 after immunization. Note: S/P = (OD450 nm of the sample − OD450 nm average of the negative control)/(OD450 nm average of the positive control − OD450 nm average of the negative control) (n = 21). * indicates a *p* < 0.05, suggesting a significant difference between the two groups. ** indicate a *p* < 0.01, suggesting a more significant difference between the two groups. *** indicate a *p* < 0.001, suggesting an extremely significant difference between the two groups.

**Figure 2 vaccines-12-01332-f002:**
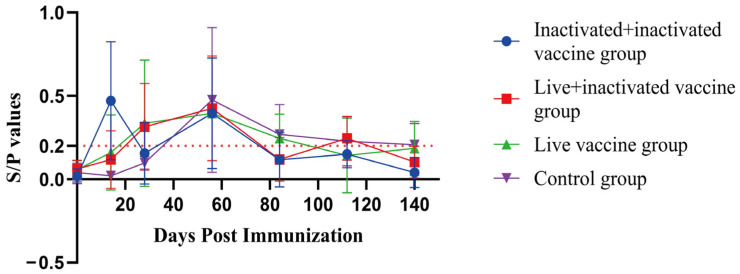
SIgA antibody growth and decline curves at different times after immunization (n = 21). Formula: S/P values = (OD450 nm of the sample − OD450 nm average of the negative control)/(OD450 nm average of the positive control − OD450 nm average of the negative control). Sample Value (S, Sample OD45 nm): the optical density (OD450 nm) value of the tested sample, which reflects the level of SIgA antibodies in the sample. Positive Control Value (P, Positive OD450 nm): the OD value of the positive control, representing a known sample containing specific antibodies (SIgA) and serving as a benchmark for the test. Negative Control Value (N, Negative OD450 nm): the OD450 nm value of the negative control, representing a sample without antibodies, used to correct for background absorbance. Purpose of the S/P Ratio: Normalizes the sample results to minimize systematic errors during the testing process. Determines whether a sample is positive and the level of antibodies, based on the S/P value.

**Figure 3 vaccines-12-01332-f003:**
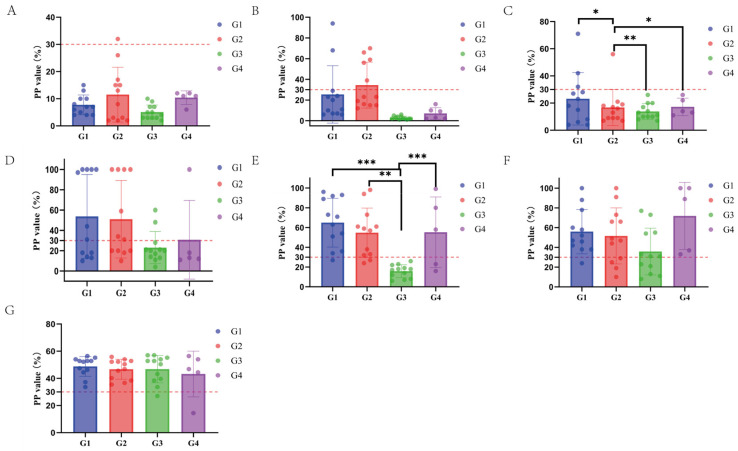
Results showing the IgG antibody levels at different times after immunization. (**A**): IgG antibody level on day 0 after immunization. (**B**): IgG antibody level on day 14 after immunization. (**C**): IgG antibody level on day 28 after immunization. (**D**): IgG antibody level on day 56 after immunization. (**E**): IgG antibody level on day 84 after immunization. (**F**): IgG antibody level on day 112 after immunization. (**G**): IgG antibody level on day 140 after immunization. Note: PP value = (OD value of the sample/OD value of the positive control) × 100% (n = 21). * indicates a *p* < 0.05, suggesting a significant difference between the two groups. ** indicate a *p* < 0.01, suggesting a more significant difference between the two groups. *** indicate a *p* < 0.001, suggesting an extremely significant difference between the two groups.

**Figure 4 vaccines-12-01332-f004:**
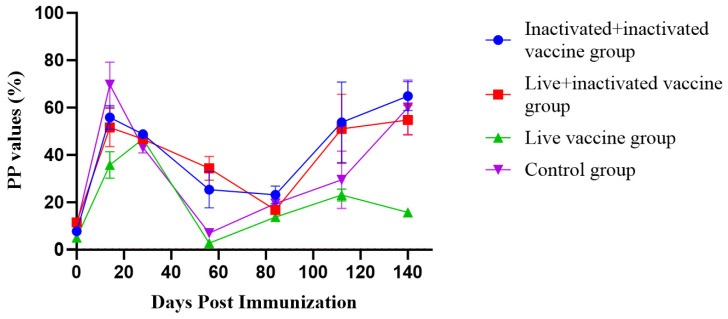
IgG antibody growth and decline curves at different times after immunization (n = 21). Definition of PP Value: PP value = (OD value of the sample/OD value of the positive control) × 100%. Meaning of the PP Value: The PP value standardizes the sample OD value, eliminating systematic errors between experimental batches and enabling more consistent comparisons. Expressed as a percentage, the PP value directly reflects the antibody level in the sample. The PP value indicates the relative reaction intensity of the sample compared to the positive control, representing the concentration or level of specific antibodies in the sample.

**Figure 5 vaccines-12-01332-f005:**
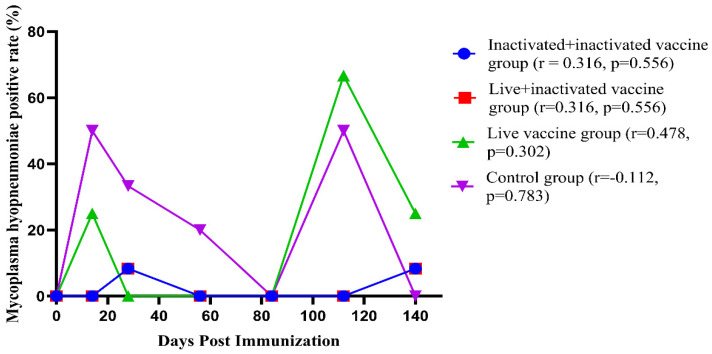
Antigen fluctuation curves at different times after immunization (n = 21). Note: the r represents the correlation coefficient between the *M. hyopneumoniae* positive rate (%) and the number of days post immunization in each group.

**Table 1 vaccines-12-01332-t001:** Field experimental design.

Groups	No. of Pigs	Age (Days)	Dosage	Vaccination Times	Immune Pathway
Group 1	98	7, 21	One (2.0 mL)	2	Intramuscular injection
Group 2	98	7, 21	One (2.0 mL)	2	Intramuscular injection (7 day) Intramuscular injection (21 day)
Group 3	96	7	One (2.0 mL)	1	Intrapulmonary injection
Group 4	21	-	-	-	-

**Table 2 vaccines-12-01332-t002:** Primer and probe sequences of *M. hyopneumoniae*.

Primer and Probe	Primer Sequence (5′–3′)	Product Length/bp
Mhp183-F	CAAAGCGAGTATGAAGAACAAGAAA	128
Mhp183-R	GTCATCATTGGGTGGCTAAGT	128
Mhp183-P	ROX-TCCAGGAAGTCAAGGTAACTAGTGACCA-BHQ	128

**Table 3 vaccines-12-01332-t003:** Results of drug use in the 4 groups during the conservation stage.

Index/Groups	Group 1	Group 2	Group 3	Group 4
Medication Usage	Florfenicol injection Lincomycin injection Sulfamonomethoxine injection Penicillin sodium Ceftiofur sodium Aminopyrine injection	Florfenicol injection Lincomycin injection Penicillin sodium Ceftiofur sodium Aminopyrine injection	Florfenicol injection Lincomycin injection	Florfenicol injection Lincomycin injection Penicillin sodium

Note: The administration route for the above treatment plan was intramuscular injection (IM). Florfenicol injection 15 mg/kg IM, twice daily, for 3 to 5 consecutive days. Lincomycin injection 10–20 mg/kg IM, twice daily, for 3 to 5 consecutive days. Sulfamonomethoxine injection 50 mg/kg IM, twice daily, for 3 to 5 consecutive days. Penicillin sodium 20,000–40,000 IU/kg IM, twice daily, for 3 to 5 consecutive days. Ceftiofur sodium 3–5 mg/kg IM, once daily, for 3 to 5 consecutive days. Aminopyrine injection 10–20 mg/kg IM, 2–3 times daily.

**Table 4 vaccines-12-01332-t004:** Results showing the weaning weight and mortality rate of suckling piglets in each group.

Index/Group	Group 1	Group 2	Group 3	Group 4
Average weaning weight (kg/pig)	5.76 ± 0.51 a	5.61 ± 0.48 b	6.63 ± 0.48 c	6.50 ± 0.65 a
Mortality rate	3.1%	1%	0	0

Note: the same letters indicate no significant difference among the groups (*p* > 0.05), and different letters indicate a significant difference among the groups (*p* < 0.05).

**Table 5 vaccines-12-01332-t005:** Results of daily weight gain and mortality rate of each group in the nursery stage.

Index/Groups	Group 1	Group 2	Group 3	Group 4
Average daily weight gain (kg/d)	0.33 ± 0.05 a	0.30 ± 0.05 b	0.35 ± 0.05 b	0.34 ± 0.06 b
Mortality rate	4.2%	2.1%	0	0

Note: the same letters indicate no significant difference among the groups (*p* > 0.05), and different letters indicate a significant difference among the groups (*p* < 0.05).

**Table 6 vaccines-12-01332-t006:** Results showing the lung slaughter score and reduction rate of *pneumonia* lesions in the fattening stage (n = 40).

Related Indicators	Group 1	Group 2	Group 3	Group 4
Consolidated lungs proportion (%)	100% (10/10)	90% (9/10)	70% (7/10)	100% (10/10)
Consolidated lungs average score	12.90 ± 4.28 a	6.78 ± 4.32 b	4.43 ± 2.44 b	14.40 ± 3.53 a
*Pneumonia* lesion reduction rate (%)	10.41%	57.64%	69.44%	/

Note: the same letters indicate no significant difference among the groups (*p* > 0.05), and different letters indicate a significant difference among the groups (*p* < 0.05).

**Table 7 vaccines-12-01332-t007:** Positive rate of the specific mucosal SIgA antibody (%) in nasal swabs and trend analysis (n = 21).

Groups	Days	Positive Rate (%)	Correlation Value	*p* Value
Group 1	0	0	−0.198	0.662
	14	66.7		
	28	33.3		
	56	58.3		
	84	16.7		
	112	8.3		
	140	8.3		
Group 2	0	0	0.179	0.713
	14	16.7		
	28	58.3		
	56	60.0		
	84	25.0		
	112	41.7		
	140	8.3		
Group 3	0	00.0	0.236	0.662
	14	25.0		
	28	58.3		
	56	58.3		
	84	41.7		
	112	8.3		
	140	41.7		
Group 4	0	0	0.694	0.110
	14	0		
	28	0		
	56	66.7		
	84	33.3		
	112	33.3		
	140	33.3		

**Table 8 vaccines-12-01332-t008:** Analysis of the positive rate (%) and trend of the specific serum IgG antibody in different age groups after immunization (n = 21).

Groups	Days	Positive Rate (%)	Correlation Value	*p* Value
Group 1	0	0.00	0.991	0.003 **
	14	16.7		
	28	25.0		
	56	58.3		
	84	91.7		
	112	91.7		
	140	100.0		
Group 2	0	8.3	0.901	0.012 *
	14	41.7		
	28	8.3		
	56	58.3		
	84	83.3		
	112	66.7		
	140	100.0		
Group 3	0	0.00	0.808	0.048 *
	14	0.00		
	28	0.00		
	56	16.7		
	84	0.00		
	112	58.3		
	140	91.7		
Group 4	0	0.0	0.946	0.007 **
	14	0.0		
	28	16.7		
	56	16.7		
	84	66.7		
	112	100.0		
	140	83.3		

Note: * indicates a *p* < 0.05, suggesting a significant difference among the groups. ** indicate a *p* < 0.01, suggesting a more significant difference among the groups.

**Table 9 vaccines-12-01332-t009:** Antigen positive rate (%) and trend analysis at different time periods after immunization (n = 21).

Groups	Days	Positive Rate (%)	Correlation Value	*p* Value
Group 1	0	0.0	0.316	0.556
	14	0.0		
	28	8.33		
	56	0.0		
	84	0.0		
	112	0.0		
	140	8.33		
Group 2	0	0.0	0.316	0.556
	14	0.0		
	28	8.33		
	56	0.0		
	84	0.0		
	112	0.0		
	140	8.33		
Group 3	0	0.0	0.478	0.302
	14	25.0		
	28	0.0		
	56	0.0		
	84	0.0		
	112	66.67		
	140	25.0		
Group 4	0	0.0	−0.112	0.783
	14	50.0		
	28	33.33		
	56	16.67		
	84	0.0		
	112	50.00		
	140	0.0		

## Data Availability

All raw data supporting the findings of this study are available by contacting the corresponding author. Data are not released because they involve the companies’ trade secrets.

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
