# Peer review of "Evaluation of the Effect of the Mycoplasma hyopneumoniae Live Vaccine (Strain 168) in Ningxiang Pigs"

_vaccines, 2024, doi:10.3390/vaccines12121332_

Round 1

Reviewer 1 Report (Previous Reviewer 3)

Comments and Suggestions for Authors

This study evaluated commercial attenuated vaccine Mycoplasma hyopneumoniae live vaccine (strain 168) in piglets from China. The manuscript provides some useful information on the use of commercial live vaccine in local piglets in China. However, minor points are required to revise as illustrated in below.

 Abstract

- Lines 20-24, details of dose and time of vaccination should be deleted, only group and received vaccine are required.

Results

- Line 255-256 in Table 6 title, author should confirm the term of trials is correctly suitable to use in this occasion. Trials mean applying the same experiment for several times. I think this word should be deleted and only keep as (n = 40) which referred to the number of samples or animals. Finally, if authors understand correctly the term trials, they should describe the number of trials not samples in each experiments if only one or more. 

- Same in lines 272, 280, 297, 301, 312, 316, and throughout the manuscript, change to (n = 21) not (n = 21 trials).

 Results and discussion

- Use styles of (* P < 0.05) with space and italic style not (*P<0.05), and 0.301 ± 0.047 not 0.301±0.047 with space between the number and its standard deviation throughout the manuscript in the text and tables, use only 2 decimal after sign (0.90 not 0.900).

Comments on the Quality of English Language

Minor revisions still required.

Author Response

Dear Editor:

Thank you very much for your and the reviews’ valuable comments on our manuscript. These comments are undoubtedly very helpful for revising and improving our manuscript. In what follows, we will address the questions you raised and provide a detailed explanation of the changes made to the original manuscript for resubmission.

The Last Submission Manuscript ID: vaccines-3311927

Comment1: Lines 20-24, details of dose and time of vaccination should be deleted, only group and received vaccine are required.

Response1: We appreciate the reviewer’s comment and have made the corresponding revisions. We have deleted details of dose and time of vaccination. (Line 21-23)

Comment 2: Line 255-256 in Table 6 title, author should confirm the term of trials is correctly suitable to use in this occasion. Trials mean applying the same experiment for several times. I think this word should be deleted and only keep as (n = 40) which referred to the number of samples or animals. Finally, if authors understand correctly the term trials, they should describe the number of trials not samples in each experiments if only one or more. 

Response 2: We appreciate the reviewer’s comment and have made the corresponding revisions. We have deleted the word “trials” in the Table 6 title into (n = 40). (Line 256)

Comment 3: Same in lines 272, 280, 297, 301, 312, 316, and throughout the manuscript, change to (n = 21) not (n = 21 trials).

Response 3: We appreciate the reviewer’s comment and have made the corresponding revisions. We have changed (n = 21 trials) to (n = 21) in the manuscript. (Line 273, 281, 298, 302, 313, 317)

Comment 4: Use styles of (* P < 0.05) with space and italic style not (*P<0.05), and 0.301 ± 0.047 not 0.301±0.047 with space between the number and its standard deviation throughout the manuscript in the text and tables, use only 2 decimal after sign (0.90 not 0.900).

Response 4: We appreciate the reviewer’s comment and have made the corresponding revisions. We have use styles of (* P < 0.05) (Line 28, 205, 232-233, 240-241, 245-246, 248, 257-258, 282-283, 303-304, 319-320, 343, 347, 356, 377, 380-381,409, 420-421, 468-469), space between the number and its standard deviation (Line 28-29, 228, 231, 235-236, 239, 256,344-345,354,378-379,469,471) and 2 decimal after sign (Line 28-29, 236-237, 240, 257, 379-380, 468,470).

Reviewer 2 Report (Previous Reviewer 1)

Comments and Suggestions for Authors

The authors have now adequately addressed my comments, and I am therefore in a position to recommend the publication of the paper in its revised form

Author Response

Dear Editor:

Thank you very much for your and the reviews’ valuable comments on our manuscript. These comments are undoubtedly very helpful for revising and improving our manuscript. Thank you again!

The Last Submission Manuscript ID: vaccines-3311927

Reviewer 3 Report (New Reviewer)

Comments and Suggestions for Authors

Infections with M. hyopneumoniae are highly prevalent in swine producing farms in different countries, and they cause significant economic losses due to decreased performance of the pigs, as well as the costs of medication use. Although antimicrobial medications do not prevent infection in pigs, it can limit the consequences of the disease and decrease the infection load. However, the widespread use of antibiotics increases the risk of antimicrobial resistance development. Vaccination against M. hyopneumoniae can be considered as a useful tool to control M. hyopneumoniae infections. However, data on the effectiveness of live vaccines are limited. Therefore, the presented manuscript is interesting and important. However some issues need to be fixed:

1. The manuscript should be checked carefully. There are errors and repetitions in the text, for example, Lines 247-249, 400.

2. Why were different regimens of antimicrobial medications used? Could the differences in daily weight gain and mortality rate be related to the different treatment regimens.

3. In my opinion, it is incorrect to use the term "antigen detection" when PCR results are presented in Section 3.7. This point should be clarified.

Author Response

Dear Editor:

Thank you very much for your and the reviews’ valuable comments on our manuscript. These comments are undoubtedly very helpful for revising and improving our manuscript. In what follows, we will address the questions you raised and provide a detailed explanation of the changes made to the original manuscript for resubmission.

The Last Submission Manuscript ID: vaccines-3311927

Comment 1: The manuscript should be checked carefully. There are errors and repetitions in the text, for example, Lines 247-249, 400.

Response 1: We appreciate the reviewer’s comment and have made the corresponding revisions. We have deleted the repetitions in the text (Line124, 245-247, 267-269, 293-294).

Comment 2: Why were different regimens of antimicrobial medications used? Could the differences in daily weight gain and mortality rate be related to the different treatment regimens.

Response 2: We appreciate the reviewer’s comment. Based on three different immunization protocols and a control group, different treatment strategies were selected during the nursery pig phase for the experimental and control groups, considering the variations in clinical symptoms and infection pressure, to achieve better therapeutic outcomes. Differences in daily weight gain and mortality rate can be related to the different treatment regimens.

Comment 3: In my opinion, it is incorrect to use the term "antigen detection" when PCR results are presented in Section 3.7. This point should be clarified.

Response 3: We appreciate the reviewer’s comment and have made the corresponding revisions. We have corrected the term "antigen detection" into “DNA detection” (Line 305).

Round 2

Reviewer 3 Report (New Reviewer)

Comments and Suggestions for Authors

The manuscript is interesting and worthy of publication.

Author Response

Dear Editor:

Thank you very much for your and the reviews’ valuable comments on our manuscript. These comments are undoubtedly very helpful for revising and improving our manuscript. In what follows, we will address the questions you raised and provide a detailed explanation of the changes made to the original manuscript for resubmission.

The Last Submission Manuscript ID: vaccines-3311927

Comment 1: The manuscript should be checked carefully. There are errors and repetitions in the text, for example, Lines 247-249, 400.

Response 1: We appreciate the reviewer’s comment and have made the corresponding revisions. We have deleted the repetitions in the text (Line124, 245-247, 267-269, 293-294).

Comment 2: Why were different regimens of antimicrobial medications used? Could the differences in daily weight gain and mortality rate be related to the different treatment regimens.

Response 2: We appreciate the reviewer’s comment. Based on three different immunization protocols and a control group, different treatment strategies were selected during the nursery pig phase for the experimental and control groups, considering the variations in clinical symptoms and infection pressure, to achieve better therapeutic outcomes. Differences in daily weight gain and mortality rate can be related to the different treatment regimens.

Comment 3: In my opinion, it is incorrect to use the term "antigen detection" when PCR results are presented in Section 3.7. This point should be clarified.

Response 3: We appreciate the reviewer’s comment and have made the corresponding revisions. We have corrected the term "antigen detection" into “DNA detection” (Line 305).

This manuscript is a resubmission of an earlier submission. The following is a list of the peer review reports and author responses from that submission.

Round 1

Reviewer 1 Report

Comments and Suggestions for Authors

The paper presents valuable data on the efficacy of vaccines against M. hyopneumoniae in pigs, but there are notable inconsistencies between the Results and Discussion sections, and certain areas lack detailed explanations. While the overall structure and methodology are sound, clearer interpretation of the findings would significantly enhance the paper's scientific rigor.

Key issues, inconsistencies, and formatting errors include:

  • Lines 2-4: The title could be revised to "Evaluation of the Efficacy of Mycoplasma hyopneumoniae Live Vaccine (Strain 168) in Ningxiang Pigs" for clarity and precision. The term "Clinical Immunization Effect" is unclear and may not be widely recognized in scientific literature.

  • Line 54: "There by" should be corrected to "thereby" as it is a single word.

  • Lines 61-65: More specific details are required about the outcomes measured in the study (e.g., morbidity, mortality, immune response), as well as the parameters used for comparison.

  • Line 82: The administration mode "intrapulmonary injection" needs a more thorough description to clarify the procedure used.

  • Line 158: The phrase "the collected data in this study was consistent with normally distributed" is grammatically incorrect. It should read "were consistent with a normal distribution."

  • Lines 122-126: The authors should provide details on the specific primers or probe sequences used for M. hyopneumoniae detection to improve transparency.

  • Table 1: The unexpected differences in the results presented in Table 1 require more in-depth explanation in the Discussion section to help readers understand their significance.

  • Lines 200-204: The reported pneumonia lesion reduction rates for Group B differ between the text (52.93%) and other sections (57.64%). These inconsistencies need to be corrected for clarity.

  • Lines 273-274: The claim that "Group A showed a higher daily weight gain compared to the other immunized groups (B and C) and the control group" contradicts the data in the Results section (Lines 172-173 and Table 1), where Group C has the highest average daily weight gain. This inconsistency should be addressed to avoid confusion.

  • Lines 312-317: The statement "there is no direct correlation between serum antibodies and immune protection" is too generalized. More specific evidence, particularly related to M. hyopneumoniae vaccination, should be provided, along with relevant references to substantiate this claim.

Reviewer 2 Report

Comments and Suggestions for Authors

An overall quite interesting study on the efficacy of a live vaccine against M hyo in pigs. An effort is presented as regards details and results of a novel vaccination protocol in a specific breed. Significant improvement is needed in materials and methods and statistical analysis parts, since the statistical power of only 10 animals tested regarding reduction of lung scores should be provided. Moreover significant performance parameters have not been measured in this study. Proper statistics are missing. Results section needs also significant improvement with addition of data on e.g. treatment plans during the course of the study. Several flaws, as demonstrated in the revision pdf file attached, do not allow acceptance of the manuscript for publication.

Comments on the Quality of English Language

English quality is quite good. A few typos or syntax issues are presented in the revision folder.

Reviewer 3 Report

Comments and Suggestions for Authors

This study evaluated commercial attenuated vaccine Mycoplasma hyopneumoniae live vaccine (strain 168) in piglets from China. The manuscript provides some useful information on the use of commercial live vaccine in local piglets in China. However, numerous limitations are occurred in this study and need to be addressed by the authors as illustrated in below.

Specific comments

Abstract and introduction

The authors should highlight the novelty points of study in both abstract briefly and in introduction in more details.

Introduction

- Information on the mechanism of successful vaccines against M. hyopneumoniae should be added based on previous literatures.

Materials and methods

- Authors should mention if the used piglets are males or females or mix.

- Details on the used experimental groups are needed and it is not clear if control group is non-vaccinated infected animals or non-vaccinated non-infected animals or vaccinated non-infected animals. Also, I recommend to prepare the experimental design in form of figure that illustrate experimental details from experimental groups, given dose and vaccination times and vaccine types, intervals and start and termination of the experiment, etc……...

- Also, number of experimental trials should be added in the text and figure legends.

- Additional experiments were importantly required for better evaluation of the used vaccines such as measuring some cytokines as IL-10, clinical findings as rectal temperature and clinical scores.

Results and discussion

- The authors should compare the obtained findings with other relevant studies.

- The authors should explain adequately the conflict in results of control groups in mortality (0%) and lung affection (high)